# Effects of Exercise Programs on Anxiety in Individuals with Disabilities: A Systematic Review with a Meta-Analysis

**DOI:** 10.3390/healthcare9081047

**Published:** 2021-08-13

**Authors:** Miguel Jacinto, Roberta Frontini, Rui Matos, Raul Antunes

**Affiliations:** 1Faculty of Sport Sciences and Physical Education, University of Coimbra, 3040-248 Coimbra, Portugal; 2Life Quality Research Centre (CIEQV), 2040-413 Rio Maior, Portugal; roberta_frontini@hotmail.com (R.F.); rui.matos@ipleiria.pt (R.M.); raul.antunes@ipleiria.pt (R.A.); 3Center for Innovative Care and Health Technology (ciTechCare), Polytechnic Institute of Leiria, 2410-541 Leiria, Portugal; 4School of Education and Social Sciences, Polytechnic Institute of Leiria, 2411-901 Leiria, Portugal

**Keywords:** anxiety, disabilities, physical exercise program

## Abstract

Anxiety symptoms are increasingly prevalent in individuals and may affect their quality of life. Physical exercise (PE) has been shown to be an effective method for reducing anxiety symptoms in the general population. The present study aimed to identify if PE programs can be a good method to reduce anxiety symptoms in individuals with disabilities, through the methodology of a systematic review with a meta-analysis. The PubMed, Web of Science, Scopus, and SPORTDiscus databases were used, considering the period from 2001 to 2021. The descriptors used were: “cerebral palsy”, “motor disability”, “physical disability”, “vision impairment”, “visual impairment”, “vision disability”, “intellectual disability”, “mental retardation”, “intellectual disabilities”, “hearing impairment”, “hearing disability”, “multiple disabilities”, “physical activity”, “exercise”, “sport”, “training”, and “anxiety”, with the Boolean operator “AND” or “OR”. The systematic review with a meta-analysis was carried out in the period between May and June 2021. The Z values (Z-values) obtained to test the null hypothesis, according to which the difference between means is zero, demonstrated a Z = 2.957, and a corresponding *p*-value of 0.003. Thus, we can reject the null hypothesis, and affirm that PE promotes positive effects and can be a good method or methodology for the reduction of anxiety symptoms of individuals with disabilities.

## 1. Introduction

Anxiety is characterized by the existence of apprehensive expectation or fear in individuals, being one of the most prevalent psychiatric symptoms across the world [1,2]. It affects approximately one out of five individuals [3,4,5], regardless of gender, race, or age groups [6]. In the International Statistical Classification of Diseases and Related Health Problems created by the World Health Organization, tenth version (ICD-10), anxiety disorders are classified in the group of mental and behavioral disorders, specifically neurotic disorders. In turn, there is a chapter on the Diagnostic and Statistical Manual of Mental Disorders [1] that contains various possible diagnoses, where anxiety disorders are presented and described.

Anxiety symptoms are also prevalent in individuals with intellectual [7], visual [8], hearing [9], and motor disabilities (including cerebral palsy) [10,11]. The presence of anxiety disorders and symptoms can affect an individual’s quality of life [12], requiring the use of health services not only because of the disease itself, but also due to the variety of its causes, including cardiovascular diseases and increased mortality risk [13,14,15,16]. 

In the population without disabilities, physical exercise (PE) has been shown to be an accessible and inexpensive option to help reduce anxiety symptoms [17,18]. PE is characterized as a planned and systematic form of physical activity, consisting of a defined structure and repetition, with the purpose of maintaining or improving one or more components of physical fitness, namely, aerobic, neuromuscular capacity, balance, and flexibility [19]. The literature reinforces that PE presents itself as an effective method for the promotion of mental health [20,21,22].

Research has shown that in undiagnosed individuals, a single session of PE can cause a reduction in anxiety symptoms [23,24]. Moreover, individuals who practice PE have a lower risk of developing anxiety disorders compared to those who do not [25,26]. Furthermore, in diagnosed individuals, PE has also proven to be an effective method in the treatment of anxiety [18,27].

Although the benefits of PE are clear and evident for the general population, namely, in the reduction anxiety symptoms, the effects in the population with disabilities are still unclear and have not been evaluated, and researchers usually direct their interests to other variables [28,29,30,31]. It is important to understand if they are transversal to the population with disabilities in order to promote their quality of life, related to the conceptual model of Schalock et al. [32], being a construct divided into three dimensions: (i) Independence, (ii) social participation, and (iii) well-being.

This is the first systematic review a with meta-analysis aimed at identifying if PE programs can be a good method to reduce anxiety symptoms in individuals with disabilities, answering the following question: Can an exercise PE program reduce anxiety symptoms in individuals with disabilities? 

## 2. Methods

### 2.1. Eligibility Criteria

This systematic review was constructed following the items of the PRISMA protocol [33] and the methodology described by Bento [34]. The protocol of this systematic review was registered in the International Prospective Register of Systematic Reviews (PROSPERO) International Prospective Register of Systematic Reviews, with registration number CRD42021256218 of 2021. The PICOS strategy [35,36] is defined as follows: (i) “P” (Patients) corresponds to participants with any type of disability, of any age, gender, ethnicity, or race; (ii) “I” (Intervention) corresponds to a PE program, implemented in the referred population, independently of the intervention time; (iii) “C” (Comparison) corresponds to the comparison before and after the intervention or between the control group and the intervention group; (iv) “O” (Outcome) corresponds to anxiety as the primary or secondary variable of focus; (v) “S” (Study Design) corresponds to intervention studies, randomized controlled trials (RCTs), or non-RCTs.

### 2.2. Information Sources and Research Strategies

The present study was carried out between May and June (Day 21) 2021, in English, by searching the databases, PubMed (all fields), Web of Science, Scopus, and SPORTDiscus (title, abstract, and keywords), considering studies from January 2001 to June 2021. The descriptors used were: “Cerebral palsy”, “motor disability”, “physical disability”, “vision impairment”, “visual impairment”, “vision disability”, “intellectual disability”, “mental retardation”, “intellectual disabilities”, “hearing impairment”, “hearing disability”, “multiple disabilities”, “physical activity”, “exercise”, “sport”, “training”, and “anxiety”, with the Boolean operator “AND” or “OR”, as shown in Table 1.

### 2.3. Inclusion Criteria

For the selection of studies, the following inclusion criteria were considered: (i) Intervention studies, RCTs, and non-RCTs; (ii) intervention studies with PE; (iii) individuals with disabilities, of the most varied types; (iv) studies with individuals of any age group, gender, race, or ethnicity.

### 2.4. Exclusion Criteria

Likewise, the following exclusion criteria were considered: (i) Studies published before 2001; (ii) studies that were not published in English or Portuguese; (iii) studies that do not describe the intervention protocol; (iv) studies in which the intervention is not just PE.

### 2.5. Data Extraction Process

The research was carried out independently by two investigators, via the ENDNOTE X7 software (Clarivate, London, United Kingdom) and duplicated articles were eliminated. In the first phase, articles were excluded based on the reading of the titles and abstract. In the second phase, which consisted of a complete reading of the articles, those that did not meet the eligibility criteria were excluded, and the study sample consisted of four articles. The results at all phases were compared by the researchers (M.J. and R.A.). One of the researchers (M.J.) exported the relevant information from the articles and inserted them into Table 2 (authorship, year of publication, country, objectives, participants, type of study, assessment instruments, duration/frequency, exercises and intensities, and main results).

### 2.6. Methodological Quality Assessment

To assess the quality of each study, the Downs and Black scale was used [37]. This scale consists of 27 items, scored with “one value” or “zero” for various parts of each article. The quality of each study was assessed by two investigators (M.J. and R.A.), independently, and they were compared and discussed to reach a consensus. When a consensus was not possible, a third investigator was available to collaborate (R.F.). The scale was divided into several score ranges, corresponding to the following quality levels: Excellent (26–28); good (20–25); fair (15–19); and poor (≤14). However, as six questions (questions 8, 11, 12, 15, 16, and 27) were not applicable to all studies, they were removed. Once modified, the scale had a maximum of 20 points compared to the original.

### 2.7. Statistical Analysis

A meta-analysis was performed using Comprehensive Meta-analysis Version 3.0 statistical software (Biostact, Inc, Englewood, United States of America). The difference in means was calculated based on information on the pre- and post-intervention means, the number of participants, and the standard deviation, using the randomized effects model to measure the effect size, with a 95% confidence interval (CI), magnitude effects, and level of statistical significance (*p* < 0.05). Heterogeneity was assessed using the chi-square, Cochran Q statistic, Higgin I squared (I^2^), and Tau square tests (T^2^). The homogeneity was verified by the asymmetry of the funnel-shaped scatter plot [38], and it was considered without publication bias when the graph had an inverted funnel [39]. 

## 3. Results

### 3.1. Selection of Studies

By searching the various databases, 330 studies were identified. In the first phase, after the elimination of duplicate articles and based on the titles and abstracts (eliminating articles that did not correspond to scientific publications, with an experimental methodology, with the implementation of a PE program, and evaluating its impact on anxiety symptoms), a sample of six studies with relevant potential for the study were identified for the next phase. Considering the eligibility criteria and the complete reading of the articles, a sample of four studies constituted the full analysis (two articles were excluded, in which the intervention was not with physical exercise or was not only with physical exercise).

Figure 1 represents a PRISMA flowchart of this systematic review.

**Table 2 healthcare-09-01047-t002:** Characteristics of the four studies.

Author, Year, Country	Aims	Participants	Type of Study	Assessment Instruments/Technique	Duration/Frequency	Exercises and Intensity
Barak et al. [40] Israel	Effects of Boccia on psychosocial outcomes in persons with severe disabilities.	*N* = 43; AA: 45.60 ± 10.95 y;multiple disabilities.	(A) Competition boccia group without professional supervision (*N* = 9); (B) professionally supervised competition bocce (*N* = 7); (C) recreational/leisure bocce (*N* = 14); and (D) control (*N* = 13).	State–Trait Anxiety Inventory; State–Trait Anxiety and Trait Anxiety Scale (36–38)—self-reports.	16 weeks; Groups A and B trained 3 × week; 90 min/session.	Groups A and B: Throwing balls to different targets; technical and tactical exercises; training and competition games. Groups A and B: Strength training (2 × week; 60 min/session). All groups participated in a rehabilitation program. Recreational participants were included in the training that emphasized tactics (2 × week), but not in the games and not in one specific training schedule.
Carraro and Gobbi [41] Italy	Investigating the effects of a 12-week exercise program on anxiety states in a group of adults with intellectual disability.	*N* = 27 (♂ = 16; ♀ = 11); AA: 40.1 ± 6.2 y;mild-to-moderate ID.	Experimental study; random groups: Training (*N* = 14) and control (*N* = 13).	Zung Self-Rating Anxiety Scale (Zung, 1971)—self-completion scale adapted for DID (Lindsay & Michie, 1988); Trace State Anxiety Inventory Form Y (STAI-Y (Spielberg, 1989)—self-completed;	12 weeks; 2 × week; 60 min/session.	Individual or paired movements using different equipment (balls, ropes, dumbbells, etc.), group cooperative situations, and adapted games.
Hardoy et al. [42] Italy	Evaluating the efficacy of an introductory mini tennis program as a therapeutic aid in the psychosocial rehabilitation of participants affected by mild-to-moderate ID.	*N* = 24; AA: 27.25 ± 8.45 y; mild ID.	Non-randomized controlled experimental study; division of groups: Training (*N* = 12) and control (*N* = 12).	Assessment and Information Rating Profile—anxiety subscale (Bouras, N. and Drummond, C. 1989)	24 weeks; 2 × week; 180 min/session.	Phase-divided PE program: 1st phase: Exercises to familiarize participants with equipment (ball, wooden paddles, and racket); 2nd phase: Development of coordination skills (oculo-manual, general dynamics, and temporal-spatial skills); 3rd phase: Learning basic tennis techniques.
Salehpoor et al. [43] Iran	Investigating the effect of exercise on the anxiety of adolescents with intellectual disabilities.	*N* = 30 ♀; A: 15–21 y; mild ID.	Quasi-experimental study; random groups: Training (*N* = 15) and control (*N* = 15).	Zung Self-Rating Anxiety Scale (1997)—self-completion.	8 weeks; 3 × week; 60 min/session.	Rhythmic aerobic exercises were performed (20 min); strength exercises (20 min)—exercise with dumbbells, ropes, and balls.

A, age; AA, average age; ID, intellectual disability; Exer, exercise/s; min, minutes; *N*, participants; y, years; ♂, male; ♀, female.

### 3.2. Origin

Two of the studies selected for full analysis were from the Asian continent [40,43] and the two others were from the European continent [41,42], with Italy being the country that has published the most studies on this subject.

### 3.3. Participants

The study of Barak [40] recruited participants with multiple disabilities, with the other three studies having a sample of individuals with intellectual disabilities only. Using an experimental methodology in all studies, there was a sum of 124 participants, 53 of whom were part of the control group. The participants were in the age group of young people and adults.

### 3.4. Assessment Instruments/Technique

All authors used a scale to assess anxiety symptoms. The scales were applied through self-reports or self-completion, with the exception of the study of Hardoy [42], where the methodology used is not clear.

### 3.5. PE Program

The PE programs were different, with no trend regarding the training methodology used. However, we can observe Boccia modality training [40], mini tennis [42], and exercises with different dynamics [41], as well as combined strength and aerobic training [43]. 

The PE focused on a modality [40,42] consisting of a first phase with exercises to familiarize the participants with the equipment and, in a second phase, with the technical and tactical drills. Barak [40] prescribed another phase with competition games, while Hardoy [40] prescribed one phase with the development of coordination skills. The central phase of the Carraro and Gobbi’s [41] and Salehpoor’s [43] studies consisted of individual or paired movements using different equipment (balls, ropes, dumbbells, etc.), group cooperative situations, and adapted games. 

The programs duration varied between 8 and 24 weeks. The weekly frequency varied between two and three times and the training sessions duration varied between 60 and 180 min. 

### 3.6. Quality of Studies

The methodological quality of the studies was assessed as poor to good. No studies were excluded due to low-quality scores. The study with the highest quality was Salehpoor’s [43], while the study with the lowest quality assessment was developed by Barak [40]. The quality ratings are shown in Table 3.

### 3.7. Results of the Interventions

Table 3 shows the results of the PE programs on anxiety symptoms in individuals with disabilities.

Taking into account the objectives of this systematic review, we found that all studies that assessed anxiety had a decrease in its (anxiety) levels, through the implementation of PE programs.

Figure 2 presents the meta-analysis results.

The sum of the effects was 1.875, which means that individuals from the intervention group presented approximately 1.9 times more probability to report improvements when compared to the control group. The range of confidence for the difference in means was 0.632 (lower limit) to 3.117 (upper limit), which means that the raw mean difference, in the universe of studies, may fall somewhere in this range. On the contrary, this range did not include a zero difference, which means that the true difference of means (true difference in means) is a value other than zero. The Z values (Z-values) obtained to test the null hypothesis, according to which the difference of means was zero, demonstrated by Z = 2.957, with a corresponding *p*-value of 0.003. Therefore, the null hypothesis can be rejected, according to which PE does not affect the anxiety of individuals with disabilities. The obtained value of Q was 69,291 with six degrees of freedom and a *p*-value <0.01. Thus, we can reject the null hypothesis that the true effect size is the same in all studies. On the contrary, the true effect size varied from study to study. In this meta-analysis, the I^2^ value obtained was 91,341, which means approximately 91% of the variance in the observed effects reflects the variance of true effects. T^2^ corresponds to the variance of the true magnitude of the effects (true effect sizes) among that studies that, in this study, presented a value of 2.482. The value of T, on the contrary, refers to the standard deviation of the true magnitude of the effects, and in the present meta-analysis equaled 1.576.

In addition, the Egger test was carried out (Figure 3), which proposes to test the null hypothesis according to which the intercept is equal to zero in the population. In Figure 2, the intercept is 1,077,080, the 95% confidence interval is (615,969, 1,538,192), with t = 6.00445 and gl = 5. The recommended *p*-value (two-tailed) is 0.00184. Thus, there is statistical evidence of the existence of publication bias. This reflects those smaller studies (which appear toward the lower) are more likely to be published if they show effects greater than the average, making them more likely to meet the criterion of statistical significance.

## 4. Discussion

This systematic review with a meta-analysis aimed to identify if PE programs can be a good method to reduce anxiety symptoms in the population with disabilities.

Intervention with a combination of factors (physical involvement, experience of skills improvement, and social relationships) [37,39] could have an anxiety-reducing effect. On the contrary, training with a modality not only promotes physical fitness and abilities such as accuracy, but also strategic planning, mental toughness, comprehensive learning processes, and social exchange [36,39]. Although the methodologies were different, socialization proved to be an important element for the success of PE programs.

It was observed that all interventions had positive effects at the level of the studied variables. In addition to the physical benefits, PE is an affordable and inexpensive option to reduce anxiety symptoms in the general population [17,18] and, as evidenced by our systematic review with a meta-analysis, in the population with disabilities.

Institutions/organizations/clubs that provide support to the target population should take into account the results of this study, namely, at the moment of planning strategies and interventions for individuals with disabilities. In addition to promoting physical fitness, PE reduces anxiety symptoms, being an asset to improving quality of life.

The present systematic review with a meta-analysis found only four studies that met the eligibility criteria, which may have limited the results and conclusions of this study. Therefore, the results should be considered with caution. At the same time, three of the four studies included only included individuals with intellectual disabilities. This fact is an indicator of the need to continue to implement PE programs among the population with disabilities, in its various types, and to understand the effect that these programs can have, not only in terms of physical health, but also mental health. Future studies should also evaluate the impact of PE on reducing anxiety symptoms in different gender or age groups. Moreover, the mechanisms involved in reducing anxiety symptoms must continue to be investigated in order to better prescribe a PE program for individuals with disabilities.

## 5. Conclusions

Taking into account the results shown in this systematic review with a meta-analysis, PE is a good method for reducing anxiety symptoms in individuals with disabilities, as well as a good method to promote their quality of life.

## Figures and Tables

**Figure 1 healthcare-09-01047-f001:**
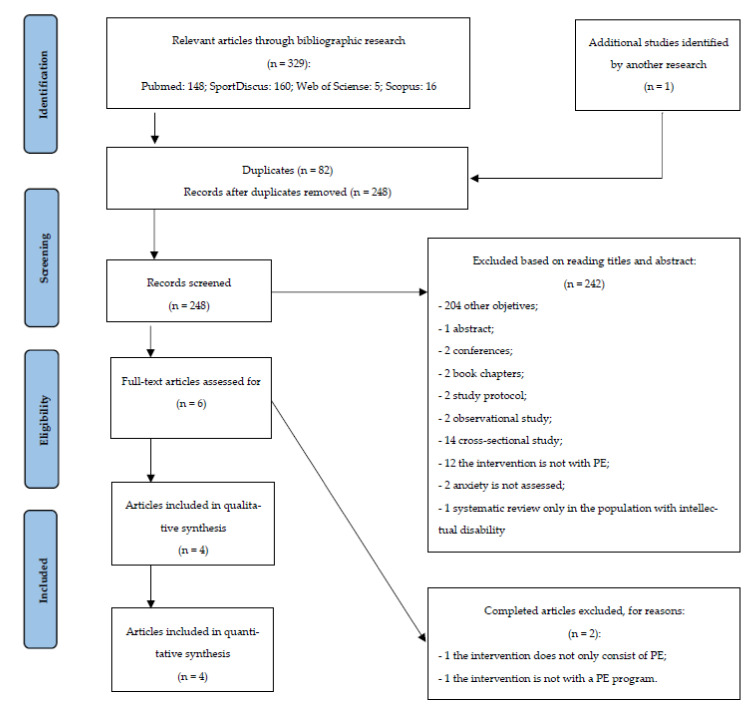
PRISMA flow diagram.

**Figure 2 healthcare-09-01047-f002:**
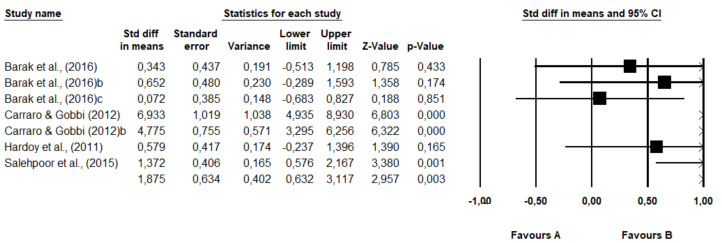
Summary of the descriptive and inferential statistics of the results of each study and the overall effect size of the effect on the anxiety symptoms in individuals with disabilities.

**Figure 3 healthcare-09-01047-f003:**
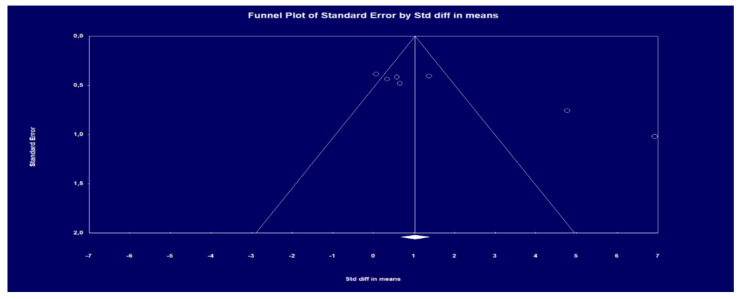
Funnel scatter plot to check publication bias.

**Table 1 healthcare-09-01047-t001:** Research strategy.

Research Number	Descriptors
1	(“brain palsy” OR “motor disability” OR “physical disability” OR “vision impairment” OR “visual impairment” OR “vision disability” OR “intellectual disability” OR “mental retardation” OR “intellectual disabilities” OR “hearing impairment” OR “hearing disability” OR “multiple disabilities”) AND (“physical activity” OR “exercise” OR sport* OR “training”) AND (“anxiety” OR “phobia” OR “panic”)

**Table 3 healthcare-09-01047-t003:** Results of the interventions on anxiety symptoms.

		Intervention Group						Control Group		Methodological Quality
		Pre-Test			Post-Test			Pre-Test	Post-Test	
Barak et al. [40]	Assessment instruments/technique	Intervention A	Intervention B	Intervention C	Intervention A	Intervention B	Intervention C	25.76 ± 10.34	25.25 ± 10.90	Poor
	Anxiety State	23.55 ± 5.38	27.00 ± 10.27	30.07 ± 4.61	20.55 ± 7.69	21.28 ± 5.82	30.07 ± 7.93			
	Trait Anxiety	24.44 ± 8.95	26.00 ± 10.68	30.50 ± 5.01	NE			25.38 ± 10.35	NE	
Carraro and Gobbi [41]	Trait Anxiety	59.9 ± 2.9			38.1 ± 2.5			59.8 ± 4.3	57.2 ± 4.3	Good
	Zung Self-Rating Anxiety Scale	33.86 ± 1.99			25.00 ± 1.62			33.46 ± 1.94	31.62 ± 1.94	
Hardoy et al. [42]	Assessment and Information Rating Profile Anxiety Subscale	3.2 ± 1.8			2.3 ± 1.3			2.8 ± 2.3	2.8 ± 2.3	Fair
Salehpoor et al. [43]	Zung’s Anxiety Scale (1997)	43.15 ± 0.96			36.60 ± 1.10			43.75 ± 1.19	45.91 ± 1.46	Good

NE, not evaluated.

## Data Availability

Additional data are available upon request to the corresponding author.

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
