# Peer review of "Effects of Exercise Programs on Anxiety in Individuals with Disabilities: A Systematic Review with a Meta-Analysis"

_healthcare, 2021, doi:10.3390/healthcare9081047_

Round 1

Reviewer 1 Report

Authors presented a very interesting meta-analysis  on the effect of exercise program on anxiety disorder in subjects affected by disabilities.

However, the document has some limitations : the authors consider together studies carried out in different age groups such as adults together with young people. I suggest to consider them separately.

As reported by the authors, the included study by Hardoy et al presented an unclear methodology on administering the anxiety symptom scale. I believe this represents a valid criterion for excluding this study from meta-analysis or the authors should contact Hardoy et al to obtain the missing information.

Author Response

Manuscript ID: healthcare-1319597

Dear Editor-in-Chief of Healthcare MDPI Journal and reviewers, We would like to thank you for the opportunity to submit a revised draft of our manuscript. The comments of the reviewers were of the utmost importance to help clarify and improve our work. We addressed all the issues raised by the reviewers to whom we wish to thank the time and effort dedicated to providing valuable feedback. We believe that at the present moment the manuscript is suitable for publication in Healthcare. All changes are highlighted in the manuscript. Below we provide a point-by-point response to the reviewers’ comments and concerns.

Response to REVIEWER 1

Comment 1: Authors presented a very interesting meta-analysis on the effect of exercise program on anxiety disorder in subjects affected by disabilities.

Response: it is motivating to receive this comment, to which we are very grateful.

Comment 2: However, the document has some limitations: the authors consider together studies carried out in different age groups such as adults together with young people. I suggest to consider them separately.

Response: Thank you very much for the comment. However, at this stage, we do not intend to separately analyse the effect of the programs on the variables of age, gender, or others, but rather to understand the current state of the art on the subject, that is, if physical exercise could be a good method to the reduction of anxiety symptoms in the general disabled population and to identify the studies that had been carried out.

Comment 3: As reported by the authors, the included study by Hardoy  et al presented an unclear methodology on administering the anxiety symptom scale. I believe this represents a valid criterion for excluding this study from meta-analysis or the authors should contact Hardoy et al to obtain the missing information.

Response: When we reported that the methodology for administering the scale to assess symptoms of anxiety was not clear, we said that we did not find information in the article on how the scale was applied. Coupled with the fact that there are few studies, we reflected that this was not a reason to exclude it and we chose to include it in our meta-analysis.

Additional clarifications

We look forward to hearing from you in due time regarding our submission and to

respond to any further questions and comments you may have.

Sincerely, Miguel Jacinto

Email: miguel.s.jacinto@ipleiria.pt

Reviewer 2 Report

GENERAL COMMENTS

The aim of this paper was to review the literature regarding the benefits of anxiety symptoms for individuals with disabilities. Although this article addresses an interesting topic, many issues should be addressed before publication.

SPECIFIC COMMENTS

ABSTRACT

I think it is a typo? Databases were reviewed until 2001? Should be 2021. Why google scholars was not used?

INTRODUCTION
The introduction needs major revision and clarification. First, the aim of the study is not clear. Review papers should answer a specific research question, which is lacking in this study. Also, the way the authors build up their introduction does not lead to the research question. Although much of the necessary information regarding the background is already written down, the authors should re-structure their introduction, explaining why their research is important. More importantly, this should lead to a clear research question.

METHODS
The methods section needs major revision. As it stands, it is not possible to replicate their study. The authors should justify why they have only searched papers from?  The authors should also add what information they extracted as well as how they analyze the data more clearly. It seems that the authors took a 3-stage process instead of 2 since after screening the title and abstracts they have looked at the full article. I am a bit confused.

RESULTS
The results section needs major revision. The authors should present information on why and how many articles were excluded (please see the Prisma). Also, the authors should add how they gathered all the information. Probably, including a research question would help the authors to structure their results. I think the authors put too much information in the tables, making the entire manuscript hard to follow.

DISCUSSION
In the discussion section, the authors should further discuss their findings and the implication of these findings. They should also discuss their findings in more depth. However, in this section, the authors present many new results. These results should be moved to the results section. The authors also discuss many topics that are not related to the results.
The limitation is also not well thought. Please revise.

The manuscript is also in need of English editing. Thank you.

Author Response

Manuscript ID: healthcare-1319597

Dear Editor-in-Chief of Healthcare MDPI Journal and reviewers, We would like to thank you for the opportunity to submit a revised draft of our manuscript. The comments of the reviewers were of the utmost importance to help clarify and improve our work. We addressed all the issues raised by the reviewers to whom we wish to thank the time and effort dedicated to providing valuable feedback. We believe that at the present moment the manuscript is suitable for publication in Healthcare. All changes are highlighted in the manuscript. Below we provide a point-by-point response to the reviewers’ comments and concerns.

Response to REVIEWER 2

GENERAL COMMENTS

The aim of this paper was to review the literature regarding the benefits of anxiety symptoms for individuals with disabilities. Although this article addresses an interesting topic, many issues should be addressed before publication.

Response: Thank you for your comment. We will take into account the recommendations mentioned and try to improve the manuscript.

SPECIFIC COMMENTS

ABSTRACT

I think it is a typo? Databases were reviewed until 2001? Should be 2021. Why google scholars was not used?

Response: Thanks for the alert. In fact, it is a bug, which has been corrected. Choosing these databases and not others was just a methodological option.

INTRODUCTION
The introduction needs major revision and clarification. First, the aim of the study is not clear. Review papers should answer a specific research question, which is lacking in this study. Also, the way the authors build up their introduction does not lead to the research question. Although much of the necessary information regarding the background is already written down, the authors should re-structure their introduction, explaining why their research is important. More importantly, this should lead to a clear research question.

Response: Thanks for the suggestions for improvement. The aims and introduction have been revised. We specifically analyzed the common thread we wanted to give to the introduction, taking advantage of all the state of the art on the topic "anxiety" already built, making the connection with exercise and population without disabilities. We tried to make the reader understand that, in the population without disabilities, physical exercise proves an important method to reducing anxiety symptoms and that, in the population with disabilities, where the prevalence of anxiety symptoms is high, a study to investigate this relationship has not yet been carried out. This fact led us to our research question: can an PE program reduce anxiety symptoms in individuals with disabilities?

METHODS
The methods section needs major revision. As it stands, it is not possible to replicate their study. The authors should justify why they have only searched papers from?  The authors should also add what information they extracted as well as how they analyze the data more clearly. It seems that the authors took a 3-stage process instead of 2 since after screening the title and abstracts they have looked at the full article. I am a bit confused.

Response: We are grateful for the comment, we tried to clarify the methods.

RESULTS
The results section needs major revision. The authors should present information on why and how many articles were excluded (please see the Prisma). Also, the authors should add how they gathered all the information. Probably, including a research question would help the authors to structure their results. I think the authors put too much information in the tables, making the entire manuscript hard to follow.

Response: The reasons for excluding the articles were added. How all the information was gathered is described in section 2.5. Data Extraction Process. Additionally, some information from table 2 has been deleted. 

DISCUSSION
In the discussion section, the authors should further discuss their findings and the implication of these findings. They should also discuss their findings in more depth. However, in this section, the authors present many new results. These results should be moved to the results section. The authors also discuss many topics that are not related to the results.
The limitation is also not well thought. Please revise.

Response: Grateful for the suggestions for improvement, which added a lot to the article. The discussion was revised to meet your suggestions.

The manuscript is also in need of English editing. Thank you.

Response: a review of English was carried out in order to avoid lapses.

Additional clarifications

We look forward to hearing from you in due time regarding our submission and to

respond to any further questions and comments you may have.

Sincerely, Miguel Jacinto

Email: miguel.s.jacinto@ipleiria.pt

Reviewer 3 Report

This study performed a systematic review to identify the benefits of physical exercise (PE) programs on anxiety symptoms in the population with disabilities and to perceive the most concrete indications for a more adequate prescription and intervention. Systematic review was performed using the Pubmed, Web of Science, Scopus and SportDiscus databases with specific descriptors in the period between 2001snf 2021. Based upon the defined inclusion and exclusion criteria, four articles were selected and then a meta-analysis was performed. Results indicated that PE promotes effects and can be a good methodology for the reduction of anxiety symptoms of individuals with disabilities.

Overall, the present manuscript focuses on important issues and employed appropriate procedures in performing the systematic review and meta-analysis. Besides, the research question focuses on an important issue involving possible effects of physical exercise and individuals with disabilities that might be of interest of Healthcare readers. Despite these positive aspects, there are several issues that need to be clarified, reviewed and improved in the manuscript. These main issues are listed below.

Major issues:

The first issue is related to the organization of the manuscript regarding the purpose and the conclusion. For instance, it would be important to change or to further discuss the following aspects:

  • the purpose presented in the abstract (lines 12-14) differs from the presented in the introduction (lines 56-58). There is the need to equalize the presentation of them;
  • moreover, the purpose at the end of the introduction seems not appropriate including that the purpose of the study was also “… perceive the most concrete indications for a more adequate prescription and intervention. Certainly, the manuscript and the procedures did not focused on this issue. This aspect is not even mentioned in the conclusion;
  • Finally, it seems that purpose is not precise in indicating clearly what the study aimed at. For instance, the study did not try “to identify the benefits of PE” on anxiety symptoms in individuals with disabilities. The study aimed to examine if PE reduces anxiety symptoms but not to identify “the benefits”. Yet, the reference to “PE programs” is also not appropriated. The study did not verify possible programs effects but PE overall;

Based upon these indications, the purpose needs to be revised and reformulated in order to appropriately indicate what the study was designed to.

The second issue is related to the justification and the discussion of the results. Most of the introduction is directed to overall anxiety. Only one mention (lines 53-55) is directed to introduce (only to introduce) the possible relation between PE and individuals with disabilities. Moreover, this mention indicates that there are studies (a few) about the thematic of the study but these studies were not presented, discussed and even referenced. Thus, the state of art about the study topic was not presented and any possible lack of knowledge, justification for the study and what the study could advance in actions are neglected. There is the need to present a thoughtful and clear revision of “these few” studies in order to readers be presented and understand the reason, rationality and the importance of the study. The same apply to the discussion that is too superficial and lack deepness in understanding and justifying the use of PE in reducing anxiety symptoms in individuals with disabilities.

Minor issues:

  • Abstract (line 15): please inform the correct period of the systematic revision state the purpose of the study;
  • Abstract (line 23): “coadjuvant” seems not appropriate;
  • Lines 92-93: please rephrase the third criterion for clarity;
  • Please include in the legend explanation (lines 2005-2006) regarding the 3 occurrences of Barak et al. (2016) and two of Carrao & Gobbi (2012) for clarity;
  • Lines 228-229: what does the identification of the existence of publication bias mean? In what does this impact the results? Such occurrence was observed for all or each study?

Conclusion (260-261): EF does not prove anything. Please clarify and further elaborate the conclusion.

Author Response

Manuscript ID: healthcare-1319597

Dear Editor-in-Chief of Healthcare MDPI Journal and reviewers, We would like to thank you for the opportunity to submit a revised draft of our manuscript. The comments of the reviewers were of the utmost importance to help clarify and improve our work. We addressed all the issues raised by the reviewers to whom we wish to thank the time and effort dedicated to providing valuable feedback. We believe that at the present moment the manuscript is suitable for publication in Healthcare. All changes are highlighted in the manuscript. Below we provide a point-by-point response to the reviewers’ comments and concerns.

Response to REVIEWER 3

This study performed a systematic review to identify the benefits of physical exercise (PE) programs on anxiety symptoms in the population with disabilities and to perceive the most concrete indications for a more adequate prescription and intervention. Systematic review was performed using the Pubmed, Web of Science, Scopus and SportDiscus databases with specific descriptors in the period between 2001snf 2021. Based upon the defined inclusion and exclusion criteria, four articles were selected and then a meta-analysis was performed. Results indicated that PE promotes effects and can be a good methodology for the reduction of anxiety symptoms of individuals with disabilities.

 Overall, the present manuscript focuses on important issues and employed appropriate procedures in performing the systematic review and meta-analysis. Besides, the research question focuses on an important issue involving possible effects of physical exercise and individuals with disabilities that might be of interest of Healthcare readers. Despite these positive aspects, there are several issues that need to be clarified, reviewed and improved in the manuscript. These main issues are listed below.

Response: Thank you for your suggestions, which we fully agree with. We tried to standardize the abstract with the introduction, reformulating our research question.

Major issues:

The first issue is related to the organization of the manuscript regarding the purpose and the conclusion. For instance, it would be important to change or to further discuss the following aspects:

  • the purpose presented in the abstract (lines 12-14) differs from the presented in the introduction (lines 56-58). There is the need to equalize the presentation of them;

Response: thank you for the suggestion. We corrected the manuscript in order to standardize the objective at different times (abstract, introduction and discussion).

  • moreover, the purpose at the end of the introduction seems not appropriate including that the purpose of the study was also “… perceive the most concrete indications for a more adequate prescription and intervention. Certainly, the manuscript and the procedures did not focused on this issue. This aspect is not even mentioned in the conclusion;

Response: We agree with your review and decided to remove the part “… perceive the most concrete indications for a more adequate prescription and intervention”

  • Finally, it seems that purpose is not precise in indicating clearly what the study aimed at. For instance, the study did not try “to identify the benefits of PE” on anxiety symptoms in individuals with disabilities. The study aimed to examine if PE reduces anxiety symptoms but not to identify “the benefits”. Yet, the reference to “PE programs” is also not appropriated. The study did not verify possible programs effects but PE overall;
  • Response: Thanks for the suggestions, we changed and clarified the aims.

Based upon these indications, the purpose needs to be revised and reformulated in order to appropriately indicate what the study was designed to.

The second issue is related to the justification and the discussion of the results. Most of the introduction is directed to overall anxiety. Only one mention (lines 53-55) is directed to introduce (only to introduce) the possible relation between PE and individuals with disabilities. Moreover, this mention indicates that there are studies (a few) about the thematic of the study but these studies were not presented, discussed and even referenced. Thus, the state of art about the study topic was not presented and any possible lack of knowledge, justification for the study and what the study could advance in actions are neglected. There is the need to present a thoughtful and clear revision of “these few” studies in order to readers be presented and understand the reason, rationality and the importance of the study. The same apply to the discussion that is too superficial and lack deepness in understanding and justifying the use of PE in reducing anxiety symptoms in individuals with disabilities.

Response: Once again, thank you very much for all your suggestions. We welcomed them and reformulated the document to meet them. Information was added in the introduction that justifies the relevance of this study, namely articles that analyze the effects of physical exercise in the population with disabilities, which lack the analysis of anxiety symptoms. At the same time, we sought to deepen the discussion, in order to justify the importance of the relationship between physical exercise-anxiety-individuals with disabilities and what future implications it may have.

Minor issues:

  • Abstract (line 15): please inform the correct period of the systematic revision state the purpose of the study;
  • Response: the correct period of the systematic revision was inserted.
  • Abstract (line 23): “coadjuvant” seems not appropriate;
  • Response: the term has been changed.
  • Lines 92-93: please rephrase the third criterion for clarity;
  • Response: the criterion was eliminated as we thought it was clear in the inclusion criteria.
  • Please include in the legend explanation (lines 2005-2006) regarding the 3 occurrences of Barak et al. (2016) and two of Carrao & Gobbi (2012) for clarity;
  • Response: the explanation was included in the legend.
  • Lines 228-229: what does the identification of the existence of publication bias mean? In what does this impact the results? Such occurrence was observed for all or each study?
  • Response: the criterion was eliminated as we thought it was clear in the inclusion criteria.

Conclusion (260-261): EF does not prove anything. Please clarify and further elaborate the conclusion.

Response: The conclusion has been clarified.

Additional clarifications

We look forward to hearing from you in due time regarding our submission and to

respond to any further questions and comments you may have.

Sincerely, Miguel Jacinto

Email: miguel.s.jacinto@ipleiria.pt

Round 2

Reviewer 1 Report

The authors have considered and careful and thoroughly changed and responded to the suggestions and considerations pointed in the previous analysis. Thanks for taking into account and accordingly changed the manuscript improving it.

Despite of all the changes, there are still a few minor aspects and changes that should be revised as indicated as follow:

Line 107: please there is the need to include a space between words

Line 259: “Most of these individuals …” please name who are “these individuals” because the reference to them occurred in the previous paragraph.

Lines 259-263: this entire paragraph, as it reads now, is confusing and awkward in both writing and context. Please revise it throughout and thoughtfully accommodating both the writing and the message that it aims to provide.

276-277: conclusion sounds better now, but: “… PE proves …” does not reflect the finding and the aim of the study. In my view, the systematic review with the meta-analysis, despite the small number of identified studies, showed that PE is a good … (PE did not prove anything, but the procedures employed showed that PE is a good method.)

278: please lower the tone regarding quality of life. Although I can envision that anxiety symptoms reduction might imply quality of life promotion, this was not the aim of the study and the results do not support such a conclusion.

Author Response

Manuscript ID: healthcare-1319597

Dear Editor-in-Chief of Healthcare MDPI Journal and reviewers, We would like to thank you for the opportunity to submit a new revised draft of our manuscript. The comments of the reviewers were of the utmost importance to help clarify and improve our work. We addressed all the issues raised by the reviewers to whom we wish to thank the time and effort dedicated to providing valuable feedback.

Response to REVIEWER 1

The authors have considered and careful and thoroughly changed and responded to the suggestions and considerations pointed in the previous analysis. Thanks for taking into account and accordingly changed the manuscript improving it. 

Despite of all the changes, there are still a few minor aspects and changes that should be revised as indicated as follow:

Comment 1: Line 107: please there is the need to include a space between words

Response: Thank you for identifying the error. The same has been fixed.

Comment 2: Line 259: “Most of these individuals …” please name who are “these individuals” because the reference to them occurred in the previous paragraph.

Response: Thanks for the comment. The names of the individuals we were referring to were added.

Comment 3: Lines 259-263: this entire paragraph, as it reads now, is confusing and awkward in both writing and context. Please revise it throughout and thoughtfully accommodating both the writing and the message that it aims to provide.

Response: We agree that the paragraph is confused. We rephrased it to make it cleaner. The massage we aim to provide is that those responsible for institutions/organizations/clubs that provide support to individuals with disabilities must take into account that PE is important to promote physical fitness, reduce anxiety symptoms and, consequently, improve their quality of life.

Comment 3: 276-277: conclusion sounds better now, but: “… PE proves …” does not reflect the finding and the aim of the study. In my view, the systematic review with the meta-analysis, despite the small number of identified studies, showed that PE is a good … (PE did not prove anything, but the procedures employed showed that PE is a good method.)

Response: Thanks for your comment. The conclusion has been changed to meet your suggestions.

Comment 3: 278: please lower the tone regarding quality of life. Although I can envision that anxiety symptoms reduction might imply quality of life promotion, this was not the aim of the study and the results do not support such a conclusion.

Response: We agree with you comment, so we have decided to delete the quality of life information from the conclusion.

Additional clarifications

We look forward to hearing from you in due time regarding our submission and to

respond to any further questions and comments you may have.

Sincerely, Miguel Jacinto

Email: miguel.s.jacinto@ipleiria.pt

Reviewer 2 Report

The authors had made significant changes as suggested. Thus, I recommend accepting the manuscript. 

Author Response

Manuscript ID: healthcare-1319597

Dear Editor-in-Chief of Healthcare MDPI Journal and reviewers,Thank you very much for your comments, we hope the documents meets your expectations.

Sincerely, Miguel Jacinto

Email: miguel.s.jacinto@ipleiria.pt